# Tracing the origin and evolution of supergene mimicry in butterflies

Wei Zhang[1], Erica Westerman[1,2], Eyal Nitzany[3], Stephanie Palmer[3] & Marcus R. Kronforst[1]

Supergene mimicry is a striking phenomenon but we know little about the evolution of this trait in any species. Here, by studying genomes of butterflies from a recent radiation in which supergene mimicry has been isolated to the gene *doublesex*, we show that sexually dimorphic mimicry and female-limited polymorphism are evolutionarily related as a result of ancient balancing selection combined with independent origins of similar morphs in different lineages and secondary loss of polymorphism in other lineages. Evolutionary loss of polymorphism appears to have resulted from an interaction between natural selection and genetic drift. Furthermore, molecular evolution of the supergene is dominated not by adaptive protein evolution or balancing selection, but by extensive hitchhiking of linked variants on the mimetic *dsx* haplotype that occurred at the origin of mimicry. Our results suggest that chance events have played important and possibly opposing roles throughout the history of this classic example of adaptation.

[1] Department of Ecology & Evolution, University of Chicago, Chicago, IL 60637, USA. [2] Department of Biological Sciences, University of Arkansas, Fayetteville, AR 72701, USA. [3] Department of Organismal Biology and Anatomy, University of Chicago, Chicago, IL 60637, USA. Correspondence and requests for materials should be addressed to M.R.K. (email: mkronforst@uchicago.edu)

The swallowtail butterfly *Papilio polytes* holds a special place in the study of mimicry. Alfred Russel Wallace, most famous for his co-discovery with Charles Darwin of evolution by natural selection[1], also made many other fundamental insights, one of which was his discovery of female-limited mimetic polymorphism in butterflies[2], a phenomenon he described in *Papilio polytes*. Then, in some of the earliest work on the genetic basis of mimicry, Fryer[3] generated experimental crosses in *Papilio polytes* to show that the Mendelian control of female polymorphism was very simple. As a result, both Punnett[4] and Fisher[5] explored the topic of *P. polytes* mimicry at length in each of their influential texts. Later, Clarke and Sheppard[6] showed that mimicry polymorphism in *P. polytes*, like a number of other female-limited polymorphic swallowtails, was controlled by just a single Mendelian locus, which they referred to as a "supergene".

Mimicry supergenes control many distinct aspects of wing patterning[6–10], as well as wing shape[11] and even behavior[12]. Furthermore, there is evidence for rare recombination among distinct elements controlled by mimicry supergenes[10]. Because of these observations, supergenes were long thought to be the product of multiple, tightly linked genes, perhaps with recombination suppressed by a chromosomal inversion polymorphism[13]. However, the mimicry supergene in *Papilio polytes* was recently characterized at a molecular genetic level and shown to be controlled by a single gene, *doublesex* (*dsx*), but with this single gene contained in an inversion polymorphism[14,15]. This discovery now opens the door to addressing long-standing questions about the evolution of supergenes and mimicry more generally[16–18].

Here we analyze whole-genome sequence data from *P. polytes* and related species and integrate these with organismal and behavioral data to (1) compare alternative hypotheses for the origin of mimicry, (2) identify the processes responsible for historical transitions between polymorphism and sexual dimorphism, (3) measure evolutionary forces acting on the *doublesex* supergene, and (4) isolate key factors that maintain long-term mimicry polymorphism. We find that the evolution of mimicry in the *P. polytes* clade has been marked by a diversity of evolutionary forces, including genetic hitchhiking at the origin of mimicry, long-term balancing selection, and even secondary loss of mimicry polymorphism in some cases. Our results suggest important roles for both selection and drift in shaping genetic and phenotypic variation across this historically significant clade of mimetic butterflies.

## Results

**Evolution of mimicry**. The *polytes* species-group of *Papilio* swallowtails consists of three species, *Papilio polytes*, *P. ambrax*, and *P. phestus*[19]. *Papilio polytes* itself consists of two strongly differentiated subspecies, *polytes* and *alphenor*, which have historically been considered either species or subspecies[6,19]. This small number of taxa encompasses a wealth of mimicry types and forms (Fig. 1). *Papilio polytes polytes*, which is broadly distributed across much of Southeast Asia, has four distinct female morphs, three of which—form *polytes*, form *theseus*, and form *romulus*—are Batesian (palatable) mimics of toxic *Pachliopta* swallowtails. The fourth female morph, form *cyrus*, is a non-mimetic form that looks similar to males. *Papilio polytes alphenor* is restricted to the Philippines and Maluku Islands and is similar to the subspecies *polytes* except that it does not display the *romulus* mimetic morph. *Papilio ambrax* and *P. phestus* are both sexually dimorphic but not polymorphic, with females displaying a mimetic wing pattern similar to the *polytes* morph of *P. polytes* and males displaying a non-mimetic color pattern. *Papilio ambrax* and *P. phestus* have adjacent, restricted distributions with

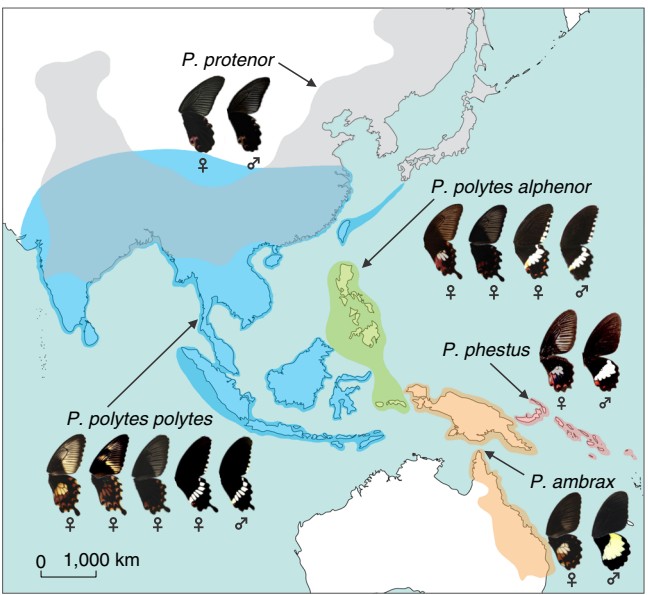

**Fig. 1** Distribution and mimicry variation of the *Papilio polytes* species-group. The *polytes* species-group of *Papilio* swallowtail butterflies consists of four historically well-characterized taxa: two differentiated subspecies within *P. polytes*, *P. polytes polytes* (female forms from left to right: *polytes* (mimetic), *romulus* (mimetic), *theseus* (mimetic), and *cyrus* (non-mimetic)); and *P. polytes alphenor* (female forms from left to right: *polytes* (mimetic), *theseus* (mimetic), and *cyrus* (non-mimetic)), plus *P. phestus* (mimetic) and *P. ambrax* (mimetic). Also pictured is the most closely related outgroup species, *P. protenor* (non-mimetic). Mimicry is highly variable across taxa, spanning sexually monomorphic, non-mimetic in *P. protenor*, female-limited mimicry polymorphism in *P. polytes polytes* and *P. polytes alphenor*, and sexually dimorphic mimicry in *P. ambrax* and *P. phestus*. Photo credit: Wei Zhang (photos and map)

*P. ambrax* located along the northeast coast of Australia, on Papua New Guinea, and surrounding islands, and *P. phestus* on the Solomon Islands and adjacent islands. The most closely related outgroup to the *polytes* species-group is *Papilio protenor*[19,20], a non-mimetic, sexually monomorphic species distributed across temperate Asia from India to Japan.

Given prior work on the genetics of mimicry in other butterflies, *Heliconius* in particular[21–23], we formulated three alternative hypotheses to explain the origin of supergene mimicry in *P. polytes* and the evolutionary relationship of mimicry among species. First, natural selection for mimicry may have resulted in multiple, independent origins of similar mimetic color patterns in different *polytes*-group lineages, a process that is expected to result in no shared genetic basis for similar phenotypes among species[24]. Convergent evolution may or may not involve the same gene or genes in different species but even if the same genes are implicated, similar phenotypes result from different mutations on different haplotypes. For example, the distantly related *Heliconius* species, *H. melpomene* and *H. erato* have converged on a diversity of nearly identical color patterns throughout their geographic range using the same suite of mimicry genes but there is no shared variation at those loci indicating independent origins of their matching color patterns[23,25]. Second, mimicry may have a shared origin among species, having arisen in one species and then been transferred to others as a result of hybridization and introgression. Such adaptive introgression of mimicry has been documented multiple times in *Heliconius*[21,26] and the white-throated sparrow supergene is suspected to have evolved this way[27]. Third, mimicry may have a shared origin among species

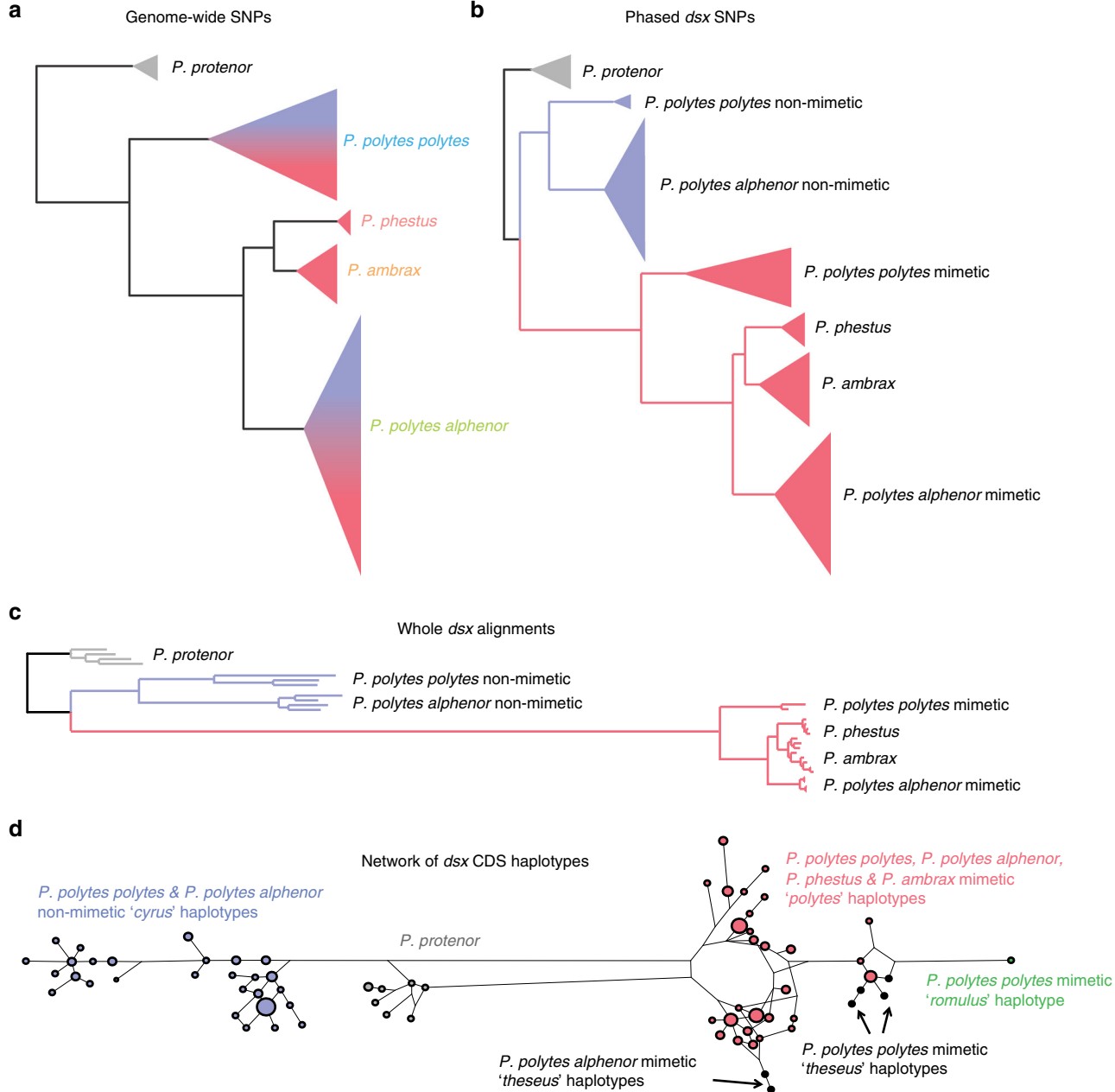

**Fig. 2** Tracing the evolution of mimicry in *Papilio*. **a** Maximum-likelihood (ML)-based phylogenetic tree of the *polytes*-group based on 40.2 million genome-wide SNPs. **b** ML tree of the *doublesex* (*dsx*) region based on 26,800 SNPs. **c** ML tree of *dsx* region based on consensus sequences (164.3 kbp, gaps included). **d** Gene network of phased *dsx* coding sequences. The blue and red colors in **a** and **b** represent non-mimetic and mimetic forms, respectively

due to shared ancestral variation. For instance, the mimetic haplotype may have arisen early in the history of the clade and been maintained as a polymorphism through successive speciation events due to balancing selection. The second two hypotheses, while both involving a shared origin for mimicry, can be distinguished by contrasting phylogenetic relationships between species and mimicry alleles, comparing times of these divergence events, as well as looking for broader signatures of hybridization and introgression among taxa using genome-scale data[21,28]. Additional open questions related to the evolution of mimicry concern how female-limited polymorphism and sexual dimorphism are evolutionarily related to one another[29–31], and if distinct mimetic morphs within *P. polytes* are related to one another or whether these represent independent origins of mimicry[6].

Our analysis of whole-genome sequencing data from 61 butterfly samples (Supplementary Table 1)—spanning all species, mimicry types, and color pattern morphs in the *polytes* clade—resulted in a well-resolved but unexpected species-level phylogeny (Fig. 2a). In contrast to prior inferences, we found that the species *Papilio polytes* was paraphyletic with respect to *P. ambrax* and *P. phestus*. This phylogeny was well supported in terms of both bootstrap support (Supplementary Fig. 1a) and fraction of the genome supporting this phylogeny (Supplementary Fig. 1b). These new phylogenetic relationships indicate a biogeographic pattern of diversification associated with dispersal from mainland Asia, across the islands of Southeast Asia, and ultimately into coastal Australia. In contrast, analysis of the 130 kbp *doublesex* inversion resulted in an entirely different tree topology, one that mapped onto mimicry, with one clade consisting of only

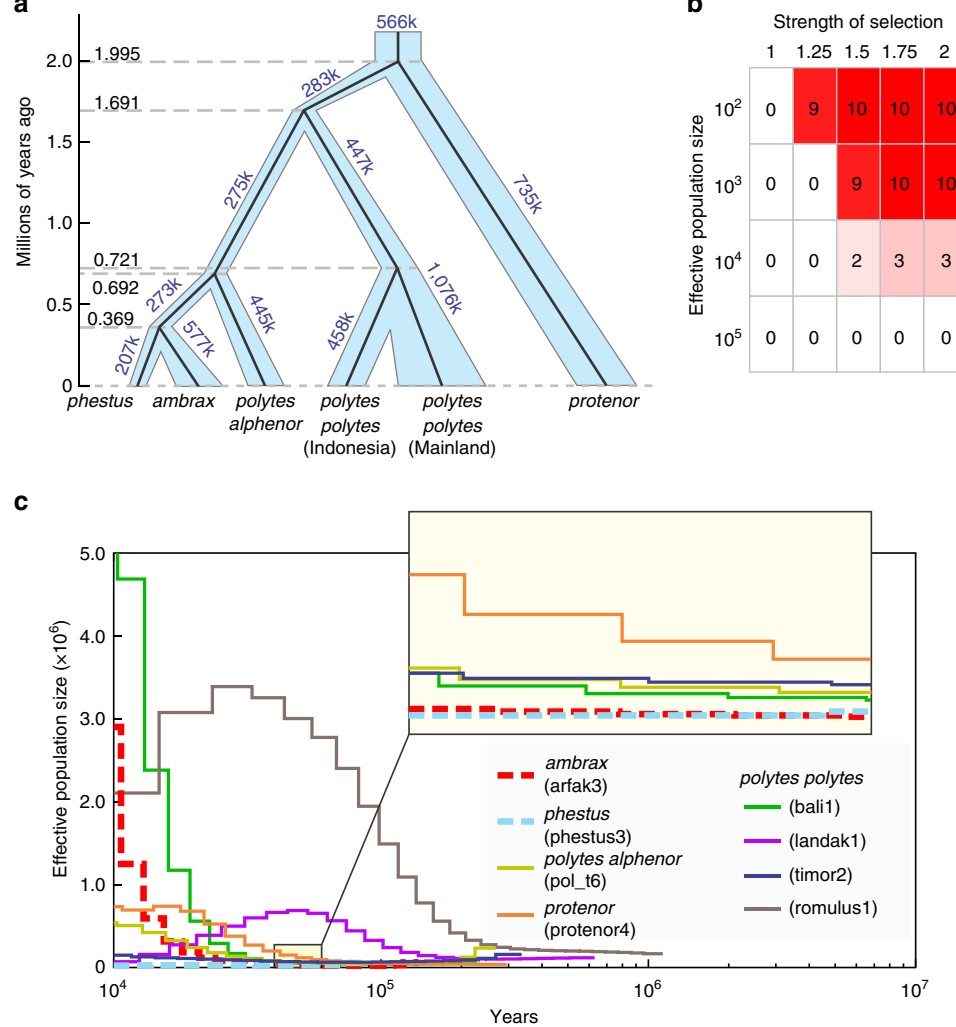

**Fig. 3** Demographic history and loss of female polymorphism. **a** Divergence times and effective population sizes were inferred among *polytes*-group taxa using G-PhoCS. Branch widths are proportional to population size and dashed line indicates divergence times. **b** Frequency of mimetic allele fixation over the course of 10 runs of a population genetic simulation incorporating varying degrees of positive selection for mimicry ($a = b = 1$, 1.25, 1.5, 1.75, or 2) and varying effective population sizes ($Ne = 10^2$, $10^3$, $10^4$, or $10^5$). All simulations were run for 400 generations and included a moderate strength of negative frequency-dependent selection ($z = 0.25$) and started at a mimetic allele frequency $= 0.25$. **c** Historical effective population sizes were inferred from eight individual genomes using PSMC, assuming a mutation rate of $\mu = 3 \times 10^{-9}$ and an average generation time of $g = 0.25$ year

non-mimetic haplotypes, from both *alphenor* and *polytes* taxa, and the other consisting of all mimetic haplotypes, spanning all species (Fig. 2b–d). A mimicry gene tree topology that groups samples by phenotype could result from either shared ancestral variation or introgression[28], but the fact that relationships among taxa within the mimetic clade of the *dsx* tree (Fig. 2b, c) are concordant with the genome-wide tree (Fig. 2a) suggests an old history of the mimetic haplotype in this radiation. Consistent with this, we inferred the timing of divergence events in the radiation (Fig. 2b) and found that the mimetic and non-mimetic haplotypes diverged prior to species-level diversification in the clade (Supplementary Fig. 2). In addition, we analyzed patterns of genetic differentiation ($F_{ST}$) across the *dsx* region and the results indicated that both *P. ambrax* and *P. phestus* share the same inversion boundaries as the *polytes* morph in *P. polytes alphenor* (Supplementary Fig. 3). Furthermore, our analyses with G-PhoCS[32] (Supplementary Fig. 2a) and Treemix[33] (Supplementary Fig. 4) yielded evidence for little gene flow among clades.

Overall, our results support a shared origin for mimicry among taxa resulting from ancestral variation. There was, however, one observation that indicated possible introgression of mimicry, the

young age estimate of divergence among mimetic haplotypes (Supplementary Fig. 2). Under a neutral model, we expect the divergence times among taxa to be similarly reflected in the two haplotypes but our results indicate that divergence between *P. polytes alphenor* and *P. polytes polytes* non-mimetic haplotypes is similar to the species tree (1.7 million years ago (Mya)), whereas divergence among mimetic haplotypes is considerably younger. This discordance is similarly reflected in estimates of synonymous divergence in the *dsx* coding sequence, which are greater among non-mimetic haplotypes than mimetic haplotypes[16]. However, the reduced sequence divergence among mimetic haplotypes is most likely a result of its peculiar evolutionary history and not introgression. For instance, a single inversion founded the pool of mimetic haplotypes and this occurred shortly before the split between the *P. polytes alphenor* and *P. polytes polytes* lineages. We suspect this greatly skewed the distribution of genetic variation between mimetic and non-mimetic haplotypes and hence estimates of divergence times. Our modeling of these different scenarios supports this (see below).

The new species-level relationships combined with the *doublesex* results, showing a single origin of mimicry, polarize

mimicry polymorphism ancestral states and indicate that the clade evolved female-limited mimicry polymorphism very early and then sexually dimorphic mimics were derived from this as a result of secondarily losing polymorphism. Interestingly, 37 years ago Vane-Wright[30] proposed a pathway model to explore patterns of evolutionary change in butterfly mimicry and he demonstrated that the evolution of sexual dimorphism with a female derived phenotype should occur via a female polymorphic ancestor. A recent analysis of *P. dardanus* and relatives has suggested that sexual dimorphism and female-limited polymorphism may not be related, but our result matches Vane-Wright's theoretical prediction exactly[30,31]. Our results also revealed that all mimetic haplotypes share a common ancestor and there have been no independent origins of mimicry in the clade. A second, independent question is whether a specific mimetic color pattern originated once or multiple times. For instance, Clarke and Sheppard[6] speculated that the dark *theseus* morph may have arisen independently multiple times. Our molecular data actually support multiple origins of the *theseus* morph because *theseus* morph haplotypes appear to be independently derived from the *polytes* morph in both the *polytes* and *alphenor* clades (Fig. 2d). Furthermore, there were no *theseus*-specific SNPs found in common between *polytes* and *alphenor* taxa across the length of the inversion, and the two groups showed a different distribution of *theseus*-specific sites across the *dsx* region (Supplementary Fig. 7).

**Secondary loss of polymorphism**. Negative frequency-dependent selection is expected to result in long-standing mimicry polymorphism[34] and our data indicate that *P. polytes polytes* and *P. polytes alphenor* have both maintained this ancient polymorphism for an estimated 2 million years (Fig. 3a). Yet, the evolutionary lineage leading to *P. ambrax* and *P. phestus* lost polymorphism by fixing the mimetic haplotype, most likely after their common ancestor diverged from *P. polytes alphenor* approximately 700,000 years ago (Fig. 3a). Because we found a geographic pattern of diversification in the clade associated with stepwise dispersal away from the Asian mainland, we were intrigued that the lineage at the periphery of the distribution was the one to have lost polymorphism. We hypothesized that demographic factors, specifically a population bottleneck associated with dispersal, may have contributed to this.

Two factors associated with reduced population size could contribute to loss of polymorphism, natural selection, and genetic drift. For instance, Charlesworth and Charlesworth[34] showed that because the fitness of a Batesian mimic is dependent on the abundance of the model species, if the number of individuals in a polymorphic mimetic species is small relative to the model, selection can rapidly fix the mimetic allele. Stochastic factors can also have a major impact on allele frequencies in small populations, particularly in the context of an expanding front where gene surfing may further amplify the effect of drift[35,36]. To explore this, we modified a population genetic model[37] to allow us to independently vary frequency-dependent selection, positive selection, and genetic drift. We simulated a variety of parameter combinations (Supplementary Fig. 8) and found that in the presence of negative frequency dependence, both drift and positive selection alone were insufficient to fix the mimetic allele under a range of parameter values but they did so in combination with high probability (Fig. 3b).

To examine the demographic history of *P. ambrax* and *P. phestus*, we used our population genomic data to infer historical effective population sizes. Overall, the results are consistent with a history of reduced population sizes in the *ambrax/phestus* lineage. For instance, although analysis with

G-PhoCS revealed no great reduction in population size at the focal internal branch, it did indicate a long history of reduced population size in this lineage, relative to others (Fig. 3a). Furthermore, analyses with PSMC[38] and MSMC[39] resulted in particularly small effective population size estimates for *P. ambrax* and *P. phestus* (Fig. 3c; Supplementary Fig. 9). Overall, our results suggest that it was the combined effects of selection and drift, likely acting during a period of reduced population size, that were critical in the evolutionary transition from female polymorphism to sexually dimorphic mimicry. Interestingly, loss of polymorphism means that today *doublesex* again simply controls sexual dimorphism in the *ambrax/phestus* lineage, but it arrived at this outcome via a remarkable history of intermediate polymorphism.

**Supergene molecular evolution**. Two processes previously implicated in the molecular evolution of the *polytes* mimicry supergene are balancing selection and adaptive protein evolution. For instance, elevated levels of polymorphism around *dsx* in *P. polytes* had been interpreted as evidence of balancing selection, in the context of frequency-dependent selection on mimicry[14]. In addition, the large number of amino acid substitutions on the mimetic haplotype had been interpreted as evidence of adaptive protein evolution associated with the evolution of mimicry[15]. However, our *dsx* SNP genealogy yielded a particularly long branch leading only to mimetic haplotypes (Fig. 2b) and this pattern was even more striking when we considered all genetic variation along the 130 kb *dsx* interval (Fig. 2c). At face value, this heavily biased sequence divergence appears inconsistent with long-term balancing selection and a simple model of coding sequence evolution.

To examine this more closely, we considered non-mimetic and mimetic haplotype groups separately and classified all nucleotide variation in the *dsx* coding sequence (CDS) as synonymous or non-synonymous, and fixed (relative to the inferred common ancestor of the mimetic and non-mimetic haplotype) or polymorphic (Fig. 4a). A classic signature of adaptive protein evolution, the foundation of the McDonald–Kreitman test[40], is an enrichment of fixed, non-synonymous variation relative to polymorphism, but there is no sign of this among mimetic haplotypes (Fisher exact test, $P = 0.112$). A comparison of non-mimetic and mimetic haplotypes revealed that they differed (Fisher $4 \times 2$ exact test, $P = 0.0004$) but this was due to elevated fixation of synonymous (Fisher exact test, $P = 0.0031$) and non-synonymous variation (Fisher exact test, $P = 0.0095$) on mimetic haplotypes. *Dsx* also showed clear signs of on-going purifying selection. For instance, most of the fixed amino acid substitutions have occurred in the middle of the gene, which is a low complexity region (LCR) of the translated protein, leaving the two functional domains of the gene, the DM domain and the dimerization domain, well-conserved (Fig. 4c). In addition, $K_a/K_s$ $(\omega) = 0.038 \pm 7.27 \times 10^{-4}$ (mean $\pm$ SE) for the non-mimetic haplotype and $0.113 \pm 3.46 \times 10^{-4}$ for the mimetic haplotype, indicating a general pattern of selective constraint on both forms of *dsx*.

We simulated the evolution of *Papilio polytes dsx* to more fully explore historical factors that may have resulted in the contrasting patterns of divergence and constraint that we see reflected in *doublesex* today. For the non-mimetic haplotype, we found a good fit to the data when simulating a large, outbred population subject to strong purifying selection ($\omega = 0.05$). This is consistent with the biology of the butterfly because this copy of *dsx* is not inverted and is associated with the ancestral, non-mimetic phenotype. For the mimetic haplotype, we considered a number of different scenarios in an effort to isolate the

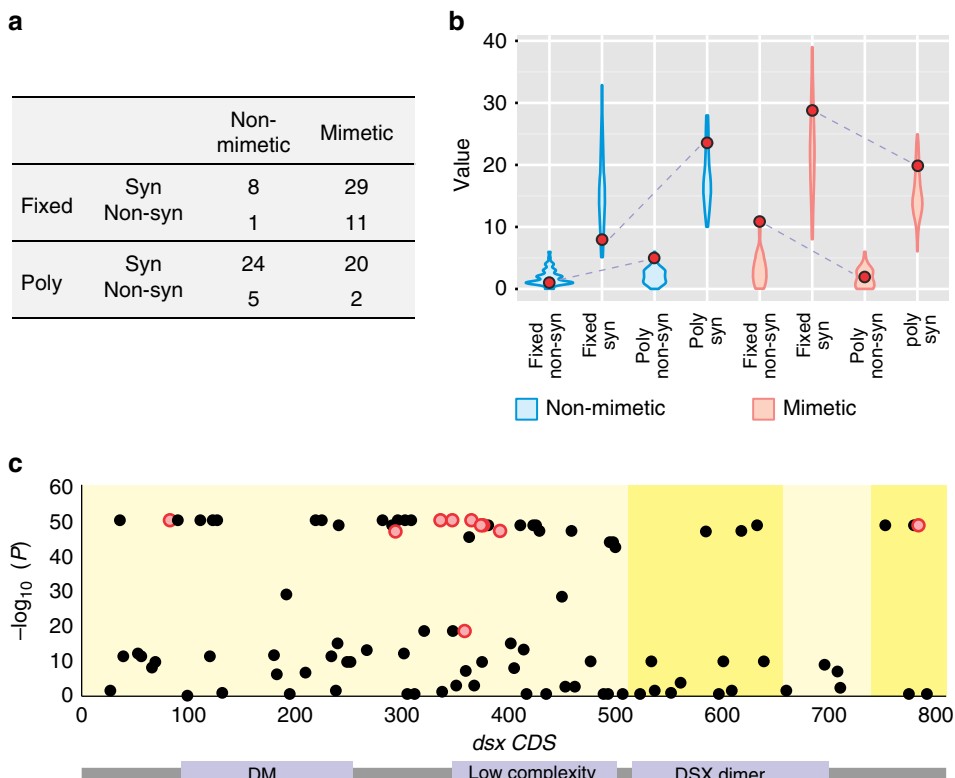

**Fig. 4** Molecular evolution of the mimicry supergene. **a** The number of fixed and polymorphic synonymous and non-synonymous substitutions in the *dsx* CDS among mimetic and non-mimetic haplotypes, in comparison to their inferred common ancestor. **b** Distributions of fixed and polymorphic synonymous and non-synonymous substitutions generated by simulations, with observed point estimates (red dots). The non-mimetic simulation involved a large population of constant size evolving under strong purifying selection ($\omega = 0.05$). The mimetic simulation contained two phases; phase 1 involved a large population of constant size evolving under strong purifying selection ($\omega = 0.05$) after which phase 2 began by randomly drawing one haplotype from the end of phase 1, continuing to evolve under strong under strong purifying selection ($\omega = 0.05$) but with a rapid increase in allele frequency. **c** SNP associations with mimicry color pattern throughout *dsx* coding region. Differential yellow shading indicates exons and functional domains of the protein are annotated below. Points are FDR-adjusted *P* values, with the fixed non-synonymous substitutions between the mimetic group and the common ancestor (from **a**) highlighted in red

factors responsible for its unique pattern of divergence (Supplementary Fig. 12). We hypothesized that three factors, in particular —positive selection, small initial starting frequency, and absence of recombination—would interact to produce the observed data. We found, however, that the pattern of enhanced fixation on the mimetic haplotype appeared solely as a consequence of genetic hitchhiking at the origin of mimicry; i.e., selecting one random haplotype from our evolving non-mimetic population to become the founder of the mimetic haplotype lineage, without changing the selection regime ($\omega = 0.05$), generated patterns of fixation and polymorphism similar to what we observe (Fig. 4b). This model also resulted in elevated sequence divergence among non-mimetic haplotypes relative to mimetic haplotypes, as previously reported[16].

Under this model, the *dsx* inversion and possibly one or a few additional mutations occurred within a short time span to generate the original mimetic genotype. The inversion also captured all additional variants present on that haplotype and "fixed" them. As the mimetic and non-mimetic copies do not appear to recombine, all of those variants remain permanently captured on the inverted copy of *dsx* to this day. If true, this would suggest that most of the substitutions in the CDS do not influence mimicry. In support of this idea, we found no non-synonymous substitutions that differed between *polytes* and *theseus* morph butterflies, in either *P. polytes polytes* or *P. polytes alphenor*. *Doublesex* is an important developmental transcription factor[41] that is otherwise highly conserved[15]. We hypothesize that

if amino acid changes on the mimetic copy of *dsx* were a product of hitchhiking and relaxed purifying selection, they may constitute a deleterious genetic load, the negative impacts of which could be far reaching. Currently, the idea that a genetic load is associated with mimicry in *P. polytes* is highly speculative but it is worth noting that supergene loci are commonly associated with a deleterious genetic load, due to both genetic hitchhiking as well as mutational decay over time[42]. Furthermore, support for this hypothesis in *P. polytes* specifically may come from Ohsaki[43], who found that mimetic females of *P. polytes polytes* did not live as long as non-mimetic females in a butterfly farm greenhouse.

**Mechanisms promoting polymorphism**. In many butterflies, mimicry polymorphisms exist between multiple mimetic morphs, and in these systems morph frequencies will stabilize where predation rates on the alternate morphs are equal[5,34]. The maintenance of a non-mimetic female morph in *Papilio polytes* is different because if there is no benefit to the non-mimetic morph, or no inherent cost to mimicry, the mimetic morph will always be equal to or better than the non-mimetic form and mimicry will fix in the population. We have proposed that a deleterious genetic load may be associated with mimicry but currently the evidence is very preliminary. However, it is worth considering this phenomenon in the context of other factors that have been proposed to explain mimicry polymorphism. To what extent

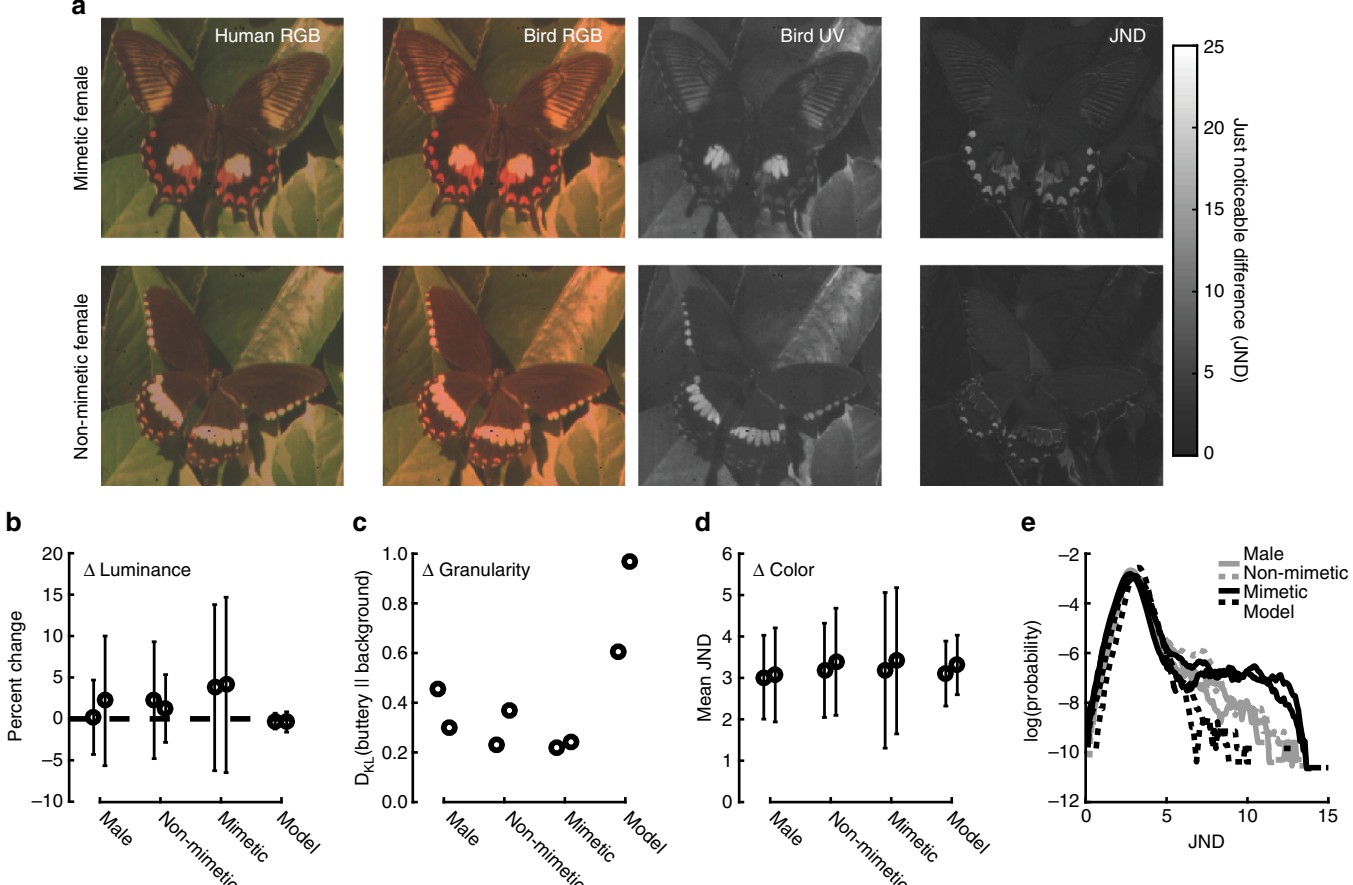

**Fig. 5** Image analysis of wing pattern coloration across morphs. **a** Two examples of a (top) mimetic "*polytes*" morph female and a (bottom) non-mimetic "*cyrus*" female. The left-most image is a standard RGB image, the next is a re-weighted RGB image balancing the red and blue channels to mimic the color sensitivities of *Leithrix lutea* vs. human cones. The next image shows the UV channel for *L. lutea*. The final image is a grayscale image of the just noticeable difference (JND) of the hyper-spectral image data, vs. the average background green, for *L. lutea*. **b** The luminance change for the butterfly vs. the background for two example individuals of each type: males, non-mimetic females, mimetic females, and two *Pachliopta aristolochiae*, the model for the mimetic morph. Error bars indicated standard deviation of percent luminance change across the butterfly. **c** The Kullback–Liebler divergence between the distribution of spatial frequencies in the image for the butterflies as compared to the background. **d** The mean and standard deviation of JND values for each butterfly. The distributions are highly non-Gaussian and are plotted on a log scale in **e**. **e** The log probability density of JND values for each individual. Photo credit: Stephanie Palmer

could a genetic load, if it existed, be the factor that offsets the benefit of mimicry?

Historically, two other mechanisms have been thought to promote polymorphism in *P. polytes*—sexual selection and crypsis—but neither has been carefully tested. First, it has been proposed that male mate preference may be tuned to the ancestral, non-mimetic phenotype, and thus sexual selection may counteract natural selection[44,45]. Westerman et al.[46] tested this across a range of populations and found something quite different; males exhibit a clear preference for mimetic females in experiments in which females do not move and males prefer active females, regardless of color pattern, in experiments in which females move. Second, it has been proposed that the evolution of mimicry may entail developing a more noticeable wing pattern and that non-mimetic females and males may be relatively cryptic[34]. We tested this using a hyper-spectral imaging camera and predator (bird) vision modeling and found that against a natural background, mimetic and non-mimetic butterflies were, in fact, equally noticeable (Fig. 5). This does not, however, account for how the butterflies might be perceived in flight. Overall, our results suggest that classic explanations for the maintenance of polymorphism in *P. polytes* may not be

responsible, opening the door to further consider the potentially subtle but widespread impacts of a deleterious genetic load.

## Discussion

From butterfly wing patterns, to self-incompatibility loci in plants, and behavioral polymorphisms in ants and birds, super-genes appear to be widespread[27,47–53]. Owing to the repeated origins of this extreme genetic architecture as a means to maintain "co-adapted" alleles, E.B. Ford[54], the father of ecological genetics stated, "…the evolution of supergenes, whether consisting of a few closely linked loci or of an inversion, should always be (but never, I think, is) treated as one of the fundamental properties of genetics." However, molecular characterization of supergenes is still rare and we know even less about their evolution. Here we have shown that chance events, from genetic hitchhiking in the formation of a supergene to allelic drift over generations, have had noticeable but opposing impacts on the long-term evolution of one iconic example of butterfly mimicry. Similarly detailed investigation of other diverse supergenes is certain to reveal novel insight into the evolutionary genetic mechanisms underlying biodiversity.

## Methods

**Sample preparation and genome re-sequencing**. Genomic DNA was extracted from 31 wild-caught adult butterflies using a phenol–chloroform DNA extraction protocol. Illumina paired-end libraries were constructed using a KAPA Hyper Prep Kit (KAPA Biosystems) and sequenced using an Illumina Hiseq2500. Raw reads were demultiplexed according to their barcodes.

**Data collection and genotype calling**. Thirty additional whole-genome re-sequencing datasets were included in this study (PRJNA234541)[14]. Low quality data with fewer than 90% bases that had a minimum quality score above 10 were filtered from raw reads and then re-sequencing data from 61 individuals were aligned to the *Papilio polytes* v1.0[15] using bowtie2-2.1.0[55] with parameter—very-sensitive-local. The mapping results were processed by re-ordering, sorting and duplicate marking in Picard v1.84 (http://broadinstitute.github.io/picard/). RealignerTargetCreator and IndelRealigner[56] in GATK 3.3 were used to realign indels and UnifiedGenotyper[57] in GATK 3.3 was used to call genotypes across 61 individuals with the following parameters: heterozygosity 0.01, stand_call_conf 50.0, stand_emit_conf 10.0, dcov 250. We also called genotypes using HaplotypeCaller[57] in GATK 3.3 and compared the two variant detection methods for calling SNPs from different species. The UnifiedGenotyper yielded more SNPs and higher mean depth per site averaged across 61 individuals for the *dsx* gene and its surrounding region. Only high-quality SNPs with Qual >30 were used for downstream analyses (Supplementary Table 1). We also simulated a set of next-generation sequencing reads to test the efficiency of our analysis pipeline. To do so, we first generated seven 1 Mb sequences using Seq-Gen.v1.3.3[58]. The relation of these sequences was set according to a maximum-likelihood phylogenetic tree (Supplementary Fig. 1a) based on our empirical data from *P. polytes polytes* (Japan), *P. polytes polytes* (Indonesia), *P. polytes polytes* (Mainland), *P. polytes alphenor*, *P. ambrax*, *P. phestus*, and *P. protenor*. The sequences were generated using a GTR model, and branch lengths were scaled using -s 0.15. Then we constructed a maximum-likelihood phylogenetic tree based on an alignment of the seven simulated sequences using RAxML[59] with the GTRGAMMA model and 100 bootstrap replicates, which resulted in a phylogeny with the same topological structure as the input. We then generated 49 Illumina paired-end in-silico libraries with different coverages (3×, 5×, 8×, 10×, 15×, 20×, and 30×) from these seven sequences using ART[60] and performed sequence alignment and genotype calling following the pipelines described above for our empirical data. According to the sequencing statistics, the mapped depth was slightly lower than simulated read depth, and the alignment rate was correlated with sequence divergence, not with read depth. With a mapped depth above 12×–15×, our pipeline was able to call a complete set of genotypes with no missing data from all the samples within *polytes* species-group, as well as from the samples in the more distantly related outgroup *P. protenor*. Our pipeline also performed well when mapped depth was lower (above 7×). Considering some of our samples had mead mapped depths as low as 3×–5×, we also tested whether low mapped depth would bias a genome-wide phylogeny. We extracted 42,367 SNPs from the simulated dataset and constructed a maximum-likelihood phylogenetic tree following the procedure described above. We did not observe any discordant pattern, and samples with different read depth were grouped according to the expected topological structure. Our results indicate that samples with different sequencing depths are suitable for downstream phylogenetic analysis using genome-wide SNPs. We also performed additional QC analyses of the data related to demographic inference and our more focused genealogical and population genetic analyses of *dsx* (see below).

**Phylogenetic analysis**. We constructed a maximum-likelihood phylogenetic tree based on genome-wide SNPs with good quality (Qual >30, ~40 million variable sites) using RAxML[59] with the GTRGAMMA model and 100 bootstrap replicates (Fig. 2a; Supplementary Fig. 1a). We also extracted 2134 non-overlapping 100 kbp windows across the genome and constructed a phylogenetic tree for each 100 kb block in the same manner. DensiTree was used to visualize 100 kb phylogenies[61] (Supplementary Fig. 1b). We constructed maximum-likelihood phylogenetic trees for the 130 kbp *dsx* region in two ways. First, we called SNPs using the mimetic *dsx* haplotype as a reference and then phased them using Beagle v4.1[62]. Then we constructed a maximum-likelihood phylogenetic tree based on the haplotype phasing results (~26,800 variable sites) using RAxML[59] with the GTRGAMMA model and 100 bootstrap replicates (Fig. 2b). Second, for each homozygous mimetic or homozygous non-mimetic individual, we called SNPs using either the mimetic or non-mimetic *dsx* haplotype as a reference and extracted a consensus sequence based on the *Papilio polytes* v1.0[15] using FastaAlternateReferenceMaker[56] in GATK 3.3. We aligned the sequences following an UCSC whole-genome alignment pipeline (http://genomewiki.ucsc.edu/index.php/Whole_genome_alignment_howto) and then extracted reciprocal-best alignment blocks using in-house scripts[63]. Then the concatenated sequences were processed using RAxML[59] with the GTRGAMMA model and 100 bootstrap replicates. The tree images were created using iTOL[64] (Fig. 2c; Supplementary Fig. 2b).

**TreeMix analysis**. We inferred migration events among populations and species using TreeMix[33] by assuming 0 to 9 migration edges (Supplementary Figs. 4–6). To

do so, we filtered out SNPs with a minor allele frequency (MAF) lower than 0.05 and estimated genome-wide allele frequency for each species or population using VCFtools[65].

**Estimating demographic parameters**. We performed genome-wide demographic inference using G-PhoCS[32], which uses multiple neutrally evolving loci throughout the genome and a Markov Chain Monte Carlo (MCMC) sampling strategy to infer demographic parameters such as population sizes, divergence times, and migration rates. The demographic analysis relies on correctly and sufficiently calling heterozygous sites, so we focus on using samples with good sequencing coverage. However, the highest coverage *phestus* sample in our analysis had a mean depth of 8.8×, so we took steps to determine the impact of read depth on our analyses. First, we tested the correlation between heterozygous calling and sequencing depth. Using 30 lab-reared *polytes* samples as an example, which should have a relatively similar genetic background, we calculated heterozygous sites for each sample using VCFtools[65] and checked the percentage of heterozygous sites per individual. The result showed a positive correlation between the number of heterozygous calls and sequencing depth but increase slowed above a sequencing depth of 8× and plateaued at 15×. Therefore, we selected six samples (PR345, protenor4, arfak3, phestus3, romulus1, and lombok1) with sufficient sequencing coverage and performed a few steps to filter the reference genome. Our filtering included selecting scaffolds with a size above N50 (3.68 Mb), masking repetitive elements using RepeatMasker and Tandem Repeats Finder, excluding conserved non-coding and 100 bp flanking regions by blasting against UCSC phastCons elements (phastcons score >0.8, size >50 bp) in the 27 way alignment for *Drosophila melanogaster*, excluding exons and 10 kb flanking regions according to the annotation of *Papilio polytes* v1.0, excluding missing calls and calls with read depth twice as high as mean depth in each sample and selecting 1 kb blocks at least 50 kb apart. We ultimately identified 1012 loci that met our criteria. The default Gamma distribution settings described by Gronau et al.[32] (mig-rate-alpha = 1.0, mig-rate-beta = 10,000, mig-rate-alpha = 0.002, mig-rate-alpha = 0.00001) were used to perform 200,000 MCMC iterations. Values were sampled every 10 iterations. The MCMC traces were viewed and evaluated manually in Tracer v1.6 (http://beast.bio.ed.ac.uk/Tracer). The raw estimates were calibrated according to Freedman et al.[66] by assuming an average mutation rate of $\mu = 3 \times 10^{-9}$[67] and an average generation time of $g = 0.25$ year. Evidence of gene flow was considered well supported if two independent analyses yielded 95% HPD lower bounds of the migration rate above zero. We conducted 15 analyses to cover each potential migration band twice and then did a full model analysis including all migration bands with significant gene flow (Fig. 3a; Supplementary Fig. 2a).

We also used PSMC[38] and MSMC[39] to infer historical effective population sizes by selecting samples with good sequencing coverage (Fig. 3c; Supplementary Fig. 9). For PSMC, the diploid consensus sequence for each sample was generated using SAMtools[68] and the sequences with ultra-low (a third of the average depth) or ultra-high (twice of the average depth) read depth were filtered out. We used an average mutation rate of $\mu = 3 \times 10^{-9}$[67] and an average generation time of $g = 0.25$ year along with the PSMC parameters set as: -N25 -t15 -r5 -p "4 + 25*2 + 4 + 6". The MSMC results were also scaled by the same mutation rate and average generation time. To test whether low sequencing depth would significantly affect demographic inference, we performed MSMC analyses for additional *ambrax* and *phestus* individuals as well (Supplementary Figs. 9 and 10). We also applied different values of uniform false negative rate (uFNR) from 0.05 to 0.5 to correct potential bias for six samples with read depth below 10× in PMSC analysis (Supplementary Fig. 11). On the basis of the results, low sequencing depth does not strongly impact the inferred demography, only slightly shifting the timing of events in the very recent past.

We estimated divergence times for the *dsx* gene using PhyTime[69], starting with the *dsx* tree topology and using the mean split time between *P. protenor* and the *polytes*-group estimated from the G-PhoCS analysis (1.995 Mya) as a calibration point (Supplementary Fig. 2b).

**Population genetic analysis**. To compare putative inversion haplotypes among populations per species, we analyzed patterns of genetic differentiation ($F_{ST}$) across the *dsx* region. $F_{ST}$ values, calculated using VCFtools[65], were compared among *P. polytes alphenor* (*cyrus* morph), *P. polytes alphenor* (*polytes* morph), *P. ambrax*, *P. phestus*, and *P. protenor* using a block size of 5 kb (Supplementary Fig. 3).

**Haplotype inference and ancestral reconstruction**. For each homozygous individual, we generated a consensus sequence of the *dsx* CDS using either the mimetic or non-mimetic *dsx* reference and FastaAlternateReferenceMaker[56] in GATK 3.3. For each heterozygous individual, we generated two consensus sequences, one with each *dsx* reference, which we then combined. We then phased these using the Haplotype Reconstruction Options implemented in DnaSP v5[70] (500 iterations and 100 burn-in). We double checked *dsx* CDS polymorphisms by PCR and Sanger sequencing using haplotype-specific and universal primers designed for exon1, exon2, and exon4, including 1_F (CTATGGTGTCCGTAGG CGC) and 1_R (GCGCCTCGTCTTGCGCCTG), 2_F (CAGGCGCAAGACGAG GC) and 2_R (GGCGCGAAGGGGGACAC), 3_F (CAGGCGCAAGACGAGGC

A) and 3_R (GGCGCGAAGGGGGACACC), 4_F (CAGGCGCAAGACGAGGC G) and 4_R (GGCGCGAAGGGGGACACA), 5_F (CAAGCGCTACTGCAAG-TAC) and 5_R (GTGCCTTCACTATAGGCGG), 6_F (CAAGCGCTACTG-CAAGTAC) and 6_R (GTGCCTTCACCATAGGCGG), 7_F (GGTGTCACTTGAAACTTTGG) and 7_R (CCTTCATCTATCTTTCGCG), 8_F (GGTGTCCCTTGAAACCTTGG) and 8_R (CCTTCATCTATCTTTCCGCG), and 9_F (GTTTGTGCGCATGTTATCGTTAC) and 9_R (GTCCACACAAAACA-CAGCACTTG), which allowed us to check SNP genotypes and correct missing calls in heterozygous individuals and homozygous individuals with lower sequencing depth. Our primers for exon 3 did not work but this small exon contains only four low-frequency synonymous SNPs (mean MAF below 0.1). We amplified *dsx* fragments from all 29 wild-caught samples, as well as pol_t6 and pol_t7. The lab-reared samples were from the same population so their CDS haplotypes could be efficiently inferred by our phasing pipeline. According to our phasing result, we summarized *dsx* genotypes in Supplementary Table 2. We also inferred the CDS of the common ancestor between mimetic and non-mimetic *dsx* haplotypes using all phased *dsx* CDS haplotypes and the software FastML[71] with the M5 model applied for codon sequences, branch lengths optimized, and marginal and joint reconstructions turned on. We inferred a network-connecting phased *dsx* CDS haplotypes using the TCS method[72] implemented in the software popart[73] (Fig. 2d).

**Local-association mapping**. Local-association mapping was performed between all the phased mimetic haplotypes and non-mimetic haplotypes across the *dsx* CDS using PLINK v1.9[74]. We calculated a P value for each SNP and then corrected it using the Behjamini–Hochberg false discovery rate (FDR) method implemented in PLINK (Fig. 4c).

**Analysis of the *theseus* morph**. The dark mimetic form *theseus* exists in both *P. polytes polytes* and *P. polytes alphenor*, but it has a patchy distribution that matches its model, a dark form of *Pachliopta aristolochiae*. The *theseus* morph generally lacks white pigmentation on the hind wing but shows variation across its distribution, with some individuals displaying small amounts of white. Clarke and Sheppard's crosses demonstrated that, like the *polytes* morph, *theseus* is dominant to *cyrus* (non-mimetic) and recessive to *romulus*, and a single brood demonstrated that *polytes* and *theseus* segregate as alleles at the mimicry supergene[6]. Clarke and Sheppard hypothesized that *polytes* and *theseus* may be controlled by similar alleles, differing only at linked and unlinked modifiers. Clarke and Sheppard also speculated that the *theseus* morph may have originated multiple times. Our results, such as the *dsx* network (Fig. 2d), support the close relationship between *theseus* and *polytes* haplotypes and the separate clusters of *theseus* haplotypes in *P. polytes polytes* and *P. polytes alphenor* are consistent with independent origins of *theseus* in the two clades (Fig. 2d).

To explore this in greater depth, we identified all fixed nucleotide differences between *theseus* and *polytes* haplotypes across the length of the *dsx* inversion, and we did this in both *P. polytes alphenor* (Philippines) and *P. polytes polytes* (Indonesia), separately. According to our *dsx* phasing result, all of the samples that displayed the *theseus* morph were *theseus/cyrus* heterozygotes, which allowed us to confidently identify *theseus* alleles at each SNP site. We then counted SNPs that were shared among *theseus* haplotypes but different from corresponding *polytes* haplotypes. For the *P. polytes alphenor* analysis, we only had two *theseus* haplotypes (pol_t6 and pol_t7) so we did not allow any missing data or alternative alleles. For the *P. polytes polytes* analysis, we used six *theseus* haplotypes (landak1, lombok1, lombok3, lombok4, jambi1, and bali1) so we allowed one missing or alternative count among them. However, since there was only one Indonesian *polytes* haplotype (timor1), we added the Japanese *polytes* haplotype, which is also *P. polytes polytes*, for comparison. In all, there were 110 *theseus*-specific SNPs in Indonesia and 107 in the Philippines. Notably, *theseus* from *P. polytes polytes* and *P. polytes alphenor* did not share any fixed SNPs and the two groups showed a different distribution theseus-specific sites across the *dsx* region (Supplementary Fig. 7). We also did not find any fixed, non-synonymous substitutions between *polytes* and *theseus* in either *P. polytes polytes* or *P. polytes alphenor*, which indicates that the difference between *theseus* and *polytes* is likely to exist outside the coding sequence.

**Population genetics simulations**. We developed a population genetic model incorporating modified negative frequency-dependent selection (NFDS) and genetic drift to explore current mimetic allele frequencies and the secondary loss of polymorphism that we found in the *ambrax/phestus* lineage. The model was built to allow the fitness (w) of a morph to be negatively correlated with its frequency (f) while also incorporating an additional benefit for the mimetic phenotype (a for homozygous mimetic genotype and b for heterozygous genotype). We borrowed a parameter (z) from Villanea et al.[37] to denote the strength of frequency-dependent selection and we also modified their R script to generate our model, which randomly sampled alleles from a finite number of alleles ($2N_e$) to incorporate stochastic variation in allele frequencies on the basis of genetic drift. The fitness of genotypes was:

$$w_{mm} = a(1 - zf_{mm} - zf_{mn}) \tag{1}$$

$$w_{nn} = 1 - zf_{nn} \tag{2}$$

$$w_{mn} = b(1 - zf_{mm} - zf_{mn}) \tag{3}$$

with m and n denoting the mimetic and non-mimetic alleles, respectively. We let z range from 0.05 to 0.95 (z = 0.05, 0.25, 0.5, 0.75, and 0.95) to model a range from neutral drift to strong frequency-dependent selection. We also tried a series of a and b sets to model pure NFDS (a = b = 1), modified NFDS with equal benefits to homozygous mimetic and heterozygous genotypes (a = b = 1.5, 3, and 5) and modified NFDS with heterozygote advantage (a = 1.5, b = 2; a = 2, b = 3; a = 4, b = 5; and a = 1.5, b = 5). We let p and q denote, respectively, the initial mimetic and non-mimetic *dsx* allele frequencies and tested a series of p and q sets (p = 0.05, q = 0.95, p = 0.25, q = 0.75, p = 0.5, q = 0.5, p = 0.75, q = 0.25; and p = 0.95, q = 0.05). The change in allele frequency relative to fitness of genotypes was calculated as:

$$p' = \frac{(p^2 w_{mm} + pq w_{mn})}{w_{mean}} \tag{4}$$

$$q' = \frac{(q^2 w_{nn} + pq w_{mn})}{w_{mean}} \tag{5}$$

Then we calculated the mean fitness as:

$$w_{mean} = p'^2 w_{mm} + 2pq w_{mn} + q'^2 w_{nn} \tag{6}$$

We also tested different population sizes ($N_e$) and a different numbers of generations. Each test included 10 separate runs. Results are shown in Supplementary Fig. 8. Comparing these results with observed genotype frequencies[46] suggests that *P. polytes* lineages are not experiencing pure NFDS (scenario 1) or forms of heterozygote advantage (scenarios 5–8) because these yield lower mimetic allele frequencies and never permit the mimicry allele fixation that we observe in the *ambrax/phestus* lineage. Very strong selection favoring the mimetic morph (scenarios 3 and 4) also seems unlikely as most populations have remained polymorphic for millions of generations. It is likely that natural systems resemble scenario 2 most closely, with a modest benefit to the mimetic morph driving higher mimicry allele frequencies while persistent NFDS maintains polymorphism over long time scales. To explore the potential interaction between genetic drift and positive selection for mimicry during the loss of polymorphism in *ambrax/phestus* (Fig. 3b), we used a moderate strength of NFDS (z = 0.25) because this resulted in a high, but never fixed, mimetic allele frequency in scenario 2.

**Molecular evolution simulations**. We simulated the evolution of *P. polytes dsx* nucleotide sequences using NetRecodon[75]. The non-mimetic *dsx* coding sequence from *Papilio polytes* v1.0[15] was used as the grand most recent common ancestor (GMRCA) sequence. A discrete gamma distribution model with three categories (K = 3) and α = 0.8 was set as the codon model. We simulated the evolution of both non-mimetic and mimetic sequences under a range of recombination rates, demographic parameters, and selection regimes (Supplementary Fig. 12). We let the recombination rate, if any, to be equal to $6 \times 10^{-6}$[76] and we used an elevated mutation rate ranging from $1.5 \times 10^{-6}$ to $3 \times 10^{-6}$ to shorten the generation time required, due to limited computing resources. We then compared resulting sequences to the GMRCA to count the number of fixed and polymorphic sites, using a MAF cutoff of 0.05 (i.e., using the same criteria as our empirical data). Five separate runs were performed for each scenario and for each run we samples 20 sequences. Simulation results were then compared with the observed counts of fixed and polymorphic synonymous and non-synonymous variation using a Fisher's exact test (Supplementary Fig. 12). An additional 50 separate runs were performed for selected scenarios highlighted in Supplementary Fig. 12 to more fully explore possible variation resulting from the best fit models (Fig. 4b).

**Color analysis**. Sixteen color channel radiance images in wavelength bands ranging from 360 to 660 nm were captured using a hyper-spectral imaging camera from Surface Optics, San Diego, CA, USA. Filming was done under outdoor, natural lighting conditions near midday. All butterflies were filmed against a leaf litter background consisting of citrus leaves from lime and lemons trees (Fig. 5). Luminance images were obtained by summing across color channels. Granularity was computed for seven equipartition spatial frequency bands up to an including the full size of the masked image[77]. The Kullback–Liebler divergence was used to quantify the difference between the butterfly and background granularity power spectra. Just noticeable difference values were computed for trichromatic human and tetrachromatic Leiothrix lutea models: we assumed a general color opponent mechanism independent of achromatic luminosity changes; we used the known spectral absorbance of each photoreceptor in each organism; we assumed constant receptor noise (valid for high intensity daylight viewing conditions); and we used standard estimates of the relative abundance of UV, short-, medium-, and long-wavelength cones in each organism[78–80].

**Code availability**. All custom codes are available from the authors upon request.

**Data availability**. The datasets generated or analyzed during this study are available from NCBI SRA (PRJNA234541 and PRJNA396246).

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

## Acknowledgements

We thank Sumitha Nallu, Roberto Marquez, Giselle Garcia, Delbert Green, Laura Southcott, and Carlos Sahagun for thoughtful discussion and Daniela Palmer for discussion and assistance assembling Fig. 2. We also thank John Novembre for advice implementing G-PhoCS and Roger Hanlon for kindly providing the hyper-spectral imager, funded by NSF Grant 1129897. This project was funded by the University of Chicago Neubauer research funds, a Pew Biomedical Scholars Fellowship, NSF grant IOS-1452648, and NIH grant GM108626 to M.R.K.

## Author contributions

W.Z. and M.R.K. designed the study; W.Z. generated genomic data, performed population genetics analyses and phylogenetics analyses, local-association tests, and population genetics and molecular evolution simulations; M.R.K. performed the network analysis; E.W., E.N., and S.P. conducted image analyses of wing-color patterns; W.Z. and M.R.K. wrote the manuscript with input from the co-authors.

## Additional information

**Competing interests:** The authors declare no competing financial interests.

