## [Peer Review File · Nature Communications]

Reviewers' comments:

Reviewer #1 (Remarks to the Author):

This article traces the evolutionary history of a supergene underlying mimetic coloration in several *Papilio* species using the genome sequences of 61 individuals. Overall, I think the authors have done a nice job combining several different analyses to form a plausible overview of the evolutionary trajectory and the demographic histories of their focal species. They have also carefully tested many of the limitations and filtering choices in their dataset with simulation. The ideas and methods developed in this study are exciting and of general interest to the readership of *Nature Communications* from my perspective. I do have some relatively minor questions and suggestions that I'd like to see addressed prior to publication:

MAF: The authors use a minor allele frequency filter set at 0.05 for several demographic analyses. This is probably a sensible cut off, but given the ongoing debates about the sensitivity of demographic analyses to truncation of the allele frequency spectrum, I would suggest that the authors justify their choice and mention potential effects of the MAF cut off. Also, could this cut off be problematic for species represented by few samples? For instance, only four *phestus* individuals were included, so missing data from one or two of these would result in removal of some *phestus*-specific alleles from the dataset.

In general, the analyses of doublesex distinguish non-mimetic and mimetic forms, but not the polymorphic mimetic forms within the two *polytes* species. The authors spend some time discussing the likelihood that the *theseus* morph arose independently in each species, and from Fig. 2d, the reader can see that there is substantial diversity among mimetic haplotypes. Maybe I missed it, but it would be interesting to know more about the relationships between the other mimetic haplotypes. For instance, in figure S3, why was this *polytes* mimetic form used? Do some morphs show greater differentiation on this scale?

I think that the demographic models investigating the loss of polymorphism in *P. ambrax* and *P. phestus* are a cool addition. I am not familiar with all analyses that the authors carried out, but I wonder whether the sampling bias, with very few *P. ambrax* and *P. phestus* individuals and (if included) a large sample of lab reared *P. polytes* individuals from a common geographic region could have biased these analyses (increased the chances of finding past genetic bottlenecks) in any way? In the methods, the authors mention that a GPhoCS analysis was performed on one individual per species, but it wasn't clear to me whether this was the main analysis that was reported, or whether it was a test of biases.

For readers without an extensive background in this system, it would be useful for the authors to mention how the supergenes are distributed in individuals with each phenotype, and in what proportion they sequenced heterozygotes. The dominance hierarchies of doublesex are briefly outlined (with reference to the *theseus* morph) only at the very end of the methods.

Finally, to the extent that the phylogenetic analyses were informed by earlier work on sex

chromosome evolution, it would be useful to point out the similarities between characterizing the evolutionary history of autosomal supergenes and sex chromosomes.

Reviewer #2 (Remarks to the Author):

This ms reports on a beautifully unexpected discovery that further stretches our understanding of the genesis of biological diversity. In reconstructing the origins of a classic mimicry polymorphism, this work provides a great example of how molecular phylogenetics can be used to test radically different alternative evolutionary hypotheses. It also challenges assumptions about the nature of supergenes and how they arise.

The core finding is the contrasting topology of genome-wide vs doublesex trees, indicating that the split between mimetic and non-mimetic forms in ancient, predating speciation within *Polytes polytes* (i.e. trans-species polymorphism). This inference is further supported by shared inversion boundaries across all mimetic forms, and weak evidence for introgression through gene flow.

I am very supportive of this effort and don't have any particular technical issues to raise; however, there are some ambiguities that it would be helpful to clarify.

1. Dsx trees group all mimetic forms by species, with incomplete information on similarities/differences between forms (e.g. use different colours on Fig 2d). What evidence is there of common origin at the level of forms (theseus discussed but not polytes explicitly)?
2. The drift x low abundance x selection hypothesis to explain the origins of *P. ambrax* and *P. phestus* is entirely plausible. What is the ecological context for this hypothesis? Principally, the abundance of alternative models for other polytes forms?
3. Does the absence of non-synonymous substitutions within dsx between theseus and polytes morphs effectively indicate that the sequence polymorphism controlling differences between these two morphs must lie outside this region? Can you speculate where?
4. Is there any indication of how dsx variation might influence colour pattern? For example, are different morph haplotypes associated with alternative splice isoforms or expression levels?
5. Partly related to query 4, the issue of how single-locus alleles apparently controlling multi-component wing pattern traits are formed, which is key to the supergene debate, is not really addressed.
6. Although there is no direct support for it, I like the genetic load hypothesis. Might it act through dosage compensation effects?

In terms of presentation:

7. The story is complex and I found myself writing form names and mimetic/non-mimetic directly onto figure 1, and returning to this at various stages. And/or you might want to consider a small 2x2 table to go with figure 1 (poly/mono x mimetic/non-mimetic).
8. Fig. 2 legend, explain blue/red colouring in a and b trees.
9. Fig. 4(a) legend, delete repeated final sentence.

10. Page 8. Are there refs to Vane-Wright's pathway model and P. dardanus analysis? This paragraph is in general a bit clunky (e.g. "...no independent origins..." followed by a superficially contradictory statement).

11. Fig S5 legend, respectively (not independently).

Ilik Saccheri

Reviewer #3 (Remarks to the Author):

The authors present a follow up to their previous study, which identified the doublesex gene as the sole loci associated with Batesian mimicry in *Papilio* butterflies. Here, the authors reconstruct the evolutionary history of this supergene and the associated mimicry trait using a thorough and well thought out population genomics study.

The study system is an elegant one, and the investigation performed here takes our understanding of this system, and more broadly of mimicry supergene evolution, a significant step forward. The manuscript is well written, enjoyable to read and relatively easy to follow. The methods appear to be sound, and most conclusions are supported by multiple lines of evidence (perhaps with the exception of some of the closing discussion on mutation load, which the authors acknowledge themselves should be considered preliminary and is included to promote future research on that point). One of the key strengths of the paper is the consideration of multiple evolutionary processes and their interaction. Overall, I greatly enjoyed this paper and I think it would be of interest to the readership of *Nature Communications*.

I have a few minor comments that the authors may want to consider.

One of the key conclusions of the paper is that functional doublesex polymorphism is ancient and existed in the shared ancestor of mimetic *P. p. polytes* and *P. p. alphenor*, maintained by balancing selection.

In the past few weeks a preprint by Guerro & Hahn <https://doi.org/10.1101/155176> makes some interesting (and I think highly relevant to this study) predictions on the behaviour of π and DXY in scenarios of sorting of previously balanced polymorphisms. I think it could be useful to consider whether the data presented here fit this model.

Page 5, lines 79-81. There are some examples of convergent evolution due to independent attained polymorphisms, for example in very distantly related species, where lineage sorting of shared ancestral polymorphisms or introgression are very unlikely, see Stern, D.L. The genetic causes of convergent evolution. *Nat. Rev. Genet.* 14, 751–764 (2013).

In the methods section for the phylogenetic analysis, it would be good to read details of how the authors considered how concatenation and incomplete lineage sorting influenced their topology.

For the TreeMix analysis, it would be useful to see the covariance matrix and have the

likelihood values for each tree with increasing numbers of m reported.

For the PSMC analysis, the authors should present the plots for the lab-reared polytes at different depths of coverage, and could use these to identify a suitable correction factor to account for the uniform false discovery of heterozygote sites in the lower coverage samples (see Figure S9.3 of Orlando et al. doi:10.1038/nature12323; and Supplementary Figure 10 of Foote et al. DOI: 10.1038/ncomms11693).

I hope these comments prove useful to the authors, and congratulate them on elegant study.

Andy Foote

Reviewers' comments:

Reviewer #1 (Remarks to the Author):

This article traces the evolutionary history of a supergene underlying mimetic coloration in several *Papilio* species using the genome sequences of 61 individuals. Overall, I think the authors have done a nice job combining several different analyses to form a plausible overview of the evolutionary trajectory and the demographic histories of their focal species. They have also carefully tested many of the limitations and filtering choices in their dataset with simulation. The ideas and methods developed in this study are exciting and of general interest to the readership of Nature Communications from my perspective. I do have some relatively minor questions and suggestions that I'd like to see addressed prior to publication:

MAF: The authors use a minor allele frequency filter set at 0.05 for several demographic analyses. This is probably a sensible cut off, but given the ongoing debates about the sensitivity of demographic analyses to truncation of the allele frequency spectrum, I would suggest that the authors justify their choice and mention potential effects of the MAF cut off. Also, could this cut off be problematic for species represented by few samples? For instance, only four *phestus* individuals were included, so missing data from one or two of these would result in removal of some *phestus*-specific alleles from the dataset.

We applied this MAF cut off for the Treemix analysis and for counting synonymous or non-synonymous sites in the *dsx* coding sequence. For the Treemix analysis, we might remove some *phestus*-specific alleles since the MAF=0.05 only allows data missed from one *phestus* sample, but this analysis was used to show little gene flow among clades, so it would tolerate data missed from a single species. For the *dsx* coding sequence, we double checked the polymorphisms by PCR and Sanger sequencing so we have minimized the data missing.

In general, the analyses of doublesex distinguish non-mimetic and mimetic forms, but not the polymorphic mimetic forms within the two *polytes* species. The authors spend some time discussing the likelihood that the *theseus* morph arose independently in each species, and from Fig. 2d, the reader can see that there is substantial diversity among mimetic haplotypes. Maybe I missed it, but it would be interesting to know more about the relationships between the other mimetic haplotypes. For instance, in figure S3, why was this *polytes* mimetic form used? Do some morphs show greater differentiation on this scale?

This is an interesting point. We were trying to trace the origin of mimicry in the *polytes* species group but didn't focus on the relationships among different mimetic morphs. We used the *polytes* morph in *P. polytes alphenor* to characterize the *dsx* inversion in our previous *dsx* paper so we used it again in Figure S3 as a "control" to indicate the same inversion boundaries in *P. ambrax* and *P. phestus*. We didn't use other morphs because all our *theseus* samples were heterozygous and we only had one *romulus* sample. Undoubtedly, it will be worth collecting more samples in future and teasing apart their relationships in detail in our follow-up study.

I think that the demographic models investigating the loss of polymorphism in *P. ambrax* and *P. phestus* are a cool addition. I am not familiar with all analyses that the authors carried out, but I wonder whether the sampling bias, with very few *P. ambrax* and *P. phestus* individuals and (if included) a large sample of lab reared *P. polytes* individuals from a common geographic region could have biased these analyses (increased the chances of finding past genetic bottlenecks) in any way? In the methods, the authors mention that a GPhoCS analysis was performed on one individual per species, but it wasn't clear to me whether this was the main analysis that was reported, or whether it was a test of biases.

GPhoCS was the main analysis. We performed PSMC, MSMC and GPhoCS analyses to investigate the loss of polymorphism in *P. ambrax* and *P. phestus*, and to perform these analyses, we only selected a sample with the best read depth for each species, so we have avoided the potential bias due to different sample size.

For readers without an extensive background in this system, it would be useful for the authors to mention how the supergenes are distributed in individuals with each phenotype, and in what proportion they sequenced heterozygotes. The dominance hierarchies of doublesex are briefly outlined (with reference to the theseus morph) only at the very end of the methods.

This is a good point. We added a supplementary table 2 to describe it in the revision.

Finally, to the extent that the phylogenetic analyses were informed by earlier work on sex chromosome evolution, it would be useful to point out the similarities between characterizing the evolutionary history of autosomal supergenes and sex chromosomes.

We agree with the reviewer's comment. In general, an autosomal supergene is defined as a cluster of linked genes that control some variable adaptive traits within a species, like sex chromosomes. But this happens slightly different to our case, because the supergene *dsx* is a single gene. So we think the mimicry supergene *dsx* is not a general case of supergene. Given this, we would prefer not mentioning sex chromosome evolution in the revision.

Reviewer #2 (Remarks to the Author):

This ms reports on a beautifully unexpected discovery that further stretches our understanding of the genesis of biological diversity. In reconstructing the origins of a classic mimicry polymorphism, this work provides a great example of how molecular phylogenetics can be used to test radically different alternative evolutionary hypotheses. It also challenges assumptions about the nature of supergenes and how they arise.

The core finding is the contrasting topology of genome-wide vs doublesex trees, indicating that the split between mimetic and non-mimetic forms in ancient, predating speciation within *Polytes polytes* (i.e.

trans-species polymorphism). This inference is further supported by shared inversion boundaries across all mimetic forms, and weak evidence for introgression through gene flow.

I am very supportive of this effort and don't have any particular technical issues to raise; however, there are some ambiguities that it would be helpful to clarify.

1. Dsx trees group all mimetic forms by species, with incomplete information on similarities/differences between forms (e.g. use different colours on Fig 2d). What evidence is there of common origin at the level of forms (theseus discussed but not polytes explicitly)?

To address this, we have used different colors to highlight different mimetic forms in Figure 2d in the revision.

2. The drift x low abundance x selection hypothesis to explain the origins of *P. ambrax* and *P. phestus* is entirely plausible. What is the ecological context for this hypothesis? Principally, the abundance of alternative models for other polytes forms?

This is a good point. We aimed to trace the origin of mimicry so we focused on comparing mimetic forms with non-mimetic forms. Currently we don't know much about the abundance of models for different *polytes* forms, but it would be a fascinating aspect and would be worth exploring in the future study.

3. Does the absence of non-synonymous substitutions within *dsx* between *theseus* and *polytes* morphs effectively indicate that the sequence polymorphism controlling differences between these two morphs must lie outside this region? Can you speculate where?

We believe that the differences between *theseus* and *polytes* morphs are not in the CDS region, but we don't have any evidence to speculate where they might be. To address this we might need to collect adequate *theseus* samples and perform association study in the future.

4. Is there any indication of how *dsx* variation might influence colour pattern? For example, are different morph haplotypes associated with alternative splice isoforms or expression levels?

We tested the expression patterns of *dsx* in *polytes* and *cyrus* forms in *P. polytes alphenor* in our previous study and we observed that some female specific *dsx* isoforms show elevated expression in *polytes* females relative to *cyrus* females in pupa stage. We didn't test the expression patterns in other mimetic forms because we were not able to collect fresh pupae of them.

5. Partly related to query 4, the issue of how single-locus alleles apparently controlling multi-component wing pattern traits are formed, which is key to the supergene debate, is not really addressed.

The single gene *dsx* is a transcription factor which gets involved in controlling sex determination and sex-limited traits via alternative splicing. We believe that different *dsx* isoforms yield different proteins that bind and regulate different downstream targets related to wing pattern traits. Now we are doing a follow-up study by performing Chip-Seq to test this hypothesis.

6. Although there is no direct support for it, I like the genetic load hypothesis. Might it act through dosage compensation effects?

This is an interesting point about maintaining the balance of deleterious mutation and selection, because dosage compensation is often related to genes on the sex chromosomes. We are not certain if the dosage compensation plays a role here but we did observe elevated expression patterns of *dsx* isoforms in mimetic females.

In terms of presentation:

7. The story is complex and I found myself writing form names and mimetic/non-mimetic directly onto figure 1, and returning to this at various stages. And/or you might want to consider a small 2x2 table to go with figure 1 (poly/mono x mimetic/non-mimetic).

We have updated the legend of figure 1 to make it clearer.

8. Fig. 2 legend, explain blue/red colouring in a and b trees.

We have explained it in the revision.

9. Fig. 4(a) legend, delete repeated final sentence.

We deleted it.

10. Page 8. Are there refs to Vane-Wright's pathway model and P. dardanus analysis? This paragraph is in general a bit clunky (e.g. "...no independent origins..." followed by a superficially contradictory statement).

We added these refs and edited the text for the paragraph on page 8.

11. Fig S5 legend, respectively (not independently).

We edited the legend.

Ilik Saccheri

Reviewer #3 (Remarks to the Author):

The authors present a follow up to their previous study, which identified the doublesex gene as the sole loci associated with Batesian mimicry in *Papilio* butterflies. Here, the authors reconstruct the evolutionary history of this supergene and the associated mimicry trait using a thorough and well thought out population genomics study.

The study system is an elegant one, and the investigation performed here takes our understanding of this system, and more broadly of mimicry supergene evolution, a significant step forward. The manuscript is well written, enjoyable to read and relatively easy to follow. The methods appear to be sound, and most conclusions are supported by multiple lines of evidence (perhaps with the exception of some of the closing discussion on mutation load, which the authors acknowledge themselves should be considered preliminary and is included to promote future research on that point). One of the key strengths of the paper is the consideration of multiple evolutionary processes and their interaction. Overall, I greatly enjoyed this paper and I think it would be of interest to the readership of *Nature Communications*.

I have a few minor comments that the authors may want to consider.

One of the key conclusions of the paper is that functional doublesex polymorphism is ancient and existed in the shared ancestor of mimetic *P. p. polytes* and *P. p. alphenor*, maintained by balancing selection.

In the past few weeks a preprint by Guerro & Hahn <https://doi.org/10.1101/155176> makes some interesting (and I think highly relevant to this study) predictions on the behaviour of *Pi* and *DXY* in scenarios of sorting of previously balanced polymorphisms. I think it could be useful to consider whether the data presented here fit this model.

This preprint makes a very interesting point, which indicates that some balanced polymorphisms might show higher divergence and might be overlooked as “islands of speciation”. However, this is a bit different from the point of our story, which mainly explains the origin of mimicry as a result of ancient balancing selection, but it is definitely worth characterizing the signature of the balancing selected *dsx* haplotypes in our future study.

Page 5, lines 79-81. There are some examples of convergent evolution due to independent attained polymorphisms, for example in very distantly related species, where lineage sorting of shared ancestral polymorphisms or introgression are very unlikely, see Stern, D.L. The genetic causes of convergent evolution. *Nat. Rev. Genet.* 14, 751–764 (2013).

We have added this reference in the revision.

In the methods section for the phylogenetic analysis, it would be good to read details of how the authors considered how concatenation and incomplete lineage sorting influenced their topology.

For the TreeMix analysis, it would be useful to see the covariance matrix and have the likelihood values for each tree with increasing numbers of m reported.

We have added the log likelihood values and the residual covariance divided by the average standard error in Supplementary Fig. S5 and Fig. S6.

For the PSMC analysis, the authors should present the plots for the lab-reared polytes at different depths of coverage, and could use these to identify a suitable correction factor to account for the uniform false discovery of heterozygote sites in the lower coverage samples (see Figure S9.3 of Orlando et al. doi:10.1038/nature12323; and Supplementary Figure 10 of Foote et al. DOI: 10.1038/ncomms11693).

We presented the plots for all the *P. ambrax* samples with different depths of coverage in Supplementary Fig. S10 using MSMC and also performed PSMC analyses for six samples with depths of coverage below 10X by applying different values of uniform False Negative Rate (uFNR) in Supplementary Fig. S11. According to these results, low depths of coverage with or without corrected by uFNR only shift the timing of events in the very recent past.

I hope these comments prove useful to the authors, and congratulate them on elegant study.
Andy Foote

REVIEWERS' COMMENTS:

Reviewer #1 (Remarks to the Author):

The authors did a wonderful job of integrating reviewer comments to improve an already excellent manuscript. I have no further comments.

Reviewer #2 (Remarks to the Author):

I am satisfied with the revisions.

Reviewer #3 (Remarks to the Author):

The authors have done a good job of addressing my comments and I feel that the paper is acceptable for publication.